# Neighborhood Walkability as a Risk Factor for Preterm Birth Phenotypes in Two Philadelphia Hospitals from 2013–2016

**DOI:** 10.3390/ijerph20115932

**Published:** 2023-05-24

**Authors:** Theresa A. Kash, Rachel F. Ledyard, Anne M. Mullin, Heather H. Burris

**Affiliations:** 1Center for Public Health Initiatives, University of Pennsylvania Perelman School of Medicine, Philadelphia, PA 19104, USA; 2Division of Neonatology, Children’s Hospital of Philadelphia, Philadelphia, PA 19104, USA; 3Tufts University School of Medicine, Boston, MA 02111, USA; 4Department of Pediatrics, University of Pennsylvania Perelman School of Medicine, Philadelphia, PA 19104, USA

**Keywords:** walkability, preterm birth, pregnancy, racial disparities

## Abstract

A total of one in ten infants is born preterm in the U.S. with large racial disparities. Recent data suggest that neighborhood exposures may play a role. Walkability—how easily individuals can walk to amenities–may encourage physical activity. We hypothesized that walkability would be associated with a decreased risk of preterm birth (PTB) and that associations would vary by PTB phenotype. PTB can be spontaneous (sPTB) from conditions such as preterm labor and preterm premature rupture of membranes, or medically indicated (mPTB) from conditions such as poor fetal growth and preeclampsia. We analyzed associations of neighborhood walkability (quantified by their Walk Score^®^ ranking) with sPTB and mPTB in a Philadelphia birth cohort (n = 19,203). Given racial residential segregation, we also examined associations in race-stratified models. Walkability (per 10 points of Walk Score ranking) was associated with decreased odds of mPTB (aOR 0.90, 95% CI: 0.83, 0.98), but not sPTB (aOR 1.04, 95% CI: 0.97, 1.12). Walkability was not protective for mPTB for all patients; there was a non-significant protective effect for White (aOR 0.87, 95% CI: 0.75, 1.01), but not Black patients (aOR 1.05, 95% CI: 0.92, 1.21) (interaction *p* = 0.03). Measuring health effects of neighborhood characteristics across populations is key for urban planning efforts focused on health equity.

## 1. Introduction

Preterm birth (PTB), or birth prior to 37 completed weeks of gestation, remains a major contributor to infant mortality and health inequity [1]. A total of 1 in 10 infants in the United Sates is born preterm, and Black infants are 50% more likely to be born preterm than White infants [2,3]. While race is associated with PTB risk, the experiences and consequences of racism—as opposed to race itself—affect complex, heterogeneous health outcomes such as PTB [3,4]. Approximately 50–60% of PTB are spontaneous (sPTB) due to conditions such as preterm labor or spontaneous rupture of membranes [5]. The remaining PTBs are medically indicated (mPTB) or initiated by a medical provider due to concern for maternal or fetal health conditions such as poor fetal growth or preeclampsia. Different pathophysiologic pathways lead to these distinct PTB phenotypes.

While several individual-level characteristics, like education, income, and self-reported race and ethnicity are associated with PTB [6], individual characteristics fail to completely account for PTB risk [7,8]. Neighborhood qualities such as walkability (i.e., how easily a person can walk to amenities) may affect PTB risk, since higher walkability may improve overall health [9]. Studies have shown that individuals who live in less walkable neighborhoods may be less physically active, and thus are more likely to develop chronic diseases such as diabetes and hypertension, which are associated with an increased risk of cardiovascular disease [9]. In a study of 23,304 patients who had singleton births across four counties in North Carolina, by Vinikoor-Imler et al., census block-level walkability was associated with a lower risk of hypertension during pregnancy and PTB among White, but not Black patients [10]. Like many PTB studies, this study did not distinguish sPTB from mPTB.

In Philadelphia, we aimed to determine whether neighborhood walkability is associated with mPTB and sPTB. As Philadelphia is an American city with longstanding residential segregation [11,12], we also examined the associations of walkability with racial disparities in PTB. We hypothesized that higher neighborhood walkability would be associated with a decreased risk of PTB, that associations would differ between sPTB and mPTB, and that racial differences in neighborhood walkability might contribute to racial disparities in PTB and its phenotypes.

## 2. Materials and Methods

### 2.1. Study Population

We performed a secondary analysis of *GeoBirth*, a retrospective cohort of patients with liveborn, singleton birth infants at the Hospital of the University of Pennsylvania (HUP) and Pennsylvania Hospital (PAH) between 23 October 2013 and 22 October 2016. *GeoBirth* is IRB-approved (University of Pennsylvania Protocol Number: 829674) to link patient data with publicly available data to identify risk factors for PTB. Patient addresses at the time of the birth hospitalization were geocoded using the “geocode address” tool on ArcMap 10.8 with the ArcGIS Street Map Premium North America version 2021.1 [13] address locator and a minimum match score of 75. The resulting X and Y coordinates were spatially joined to the US Census Bureau’s 2019 cartographic boundary shapefile [14] to assign each address a census tract. We limited our cohort to patients who lived within the Philadelphia census tract.

### 2.2. Exposure—Walkability

Walkability was measured using a Walk Score ranking (range 0–100) from WalkScore.com (accessed on 1 March 2021) [15], a website that determines walkability by how easily a person can walk to nearby amenities [16]. The higher the Walk Score ranking, the more walkable a neighborhood is (0–49 is car-dependent, 50–69 is somewhat walkable, 70–89 is very walkable, and 90–100 is “walker’s paradise”). The website uses several variables to assign a Walk Score ranking for each neighborhood, including walking routes, nearby amenities, and pedestrian friendliness. The Walk Score ranking is calculated using a point system, where points are determined by distance to amenities and the ability to conduct daily errands. For example, 0.25 points are awarded per amenity that is within a five-minute walk. A decay function is used to assign points to amenities further away; no points are awarded after a 30-min walk distance. Pedestrian friendliness is analyzed by looking at population density and various road metrics like block length and intersection density [16]. Walkability for Philadelphia was determined by searching “Philadelphia, Pennsylvania” on the website. The Walk Score ranking average was available for 77 neighborhoods in Philadelphia [15]. We spatially joined the WalkScore.com neighborhood polygons outlined by Google Maps to our dataset using the NAD 1983 StatePlane Pennsylvania South FIPS 3702 (US Feet) projected coordinate system (Figure 1) using ArcMap 10.8.1, Esri Corporation [13,17].

### 2.3. Outcome—PTB, sPTB and mPTB

PTB was defined as birth between 20 0/7 and 36 6/7 weeks of gestation and was dichotomized as sPTB or mPTB [18]. We used the best obstetric estimate of gestational age assigned by obstetric providers using a combination of the last menstrual period (if known) and the earliest ultrasound [19]. The two types of PTB are sPTB, due to conditions such as preterm labor or spontaneous rupture of membranes [5], and mPTB or PTB, initiated by a medical provider due to concern for maternal or fetal health conditions, such as poor fetal growth or preeclampsia. A total of two independent, blinded reviewers classified each PTB as either sPTB or mPTB by reading the clinical notes from the inpatient admission for labor and delivery. When there was non-concordance, a third reviewer adjudicated the PTB. If the birth remained difficult to classify, it went to a neonatologist (HHB) for final review.

### 2.4. Covariate Ascertainment

Demographic characteristics included maternal age (<25; 25-<35; and ≥35 years), maternal race and ethnicity (Non-Hispanic White, Non-Hispanic Black, Asian, Hispanic, multiracial, other, unknown), insurance type (public compared to private), pre-pregnancy body mass index (BMI) (<25; 25-<30; and ≥30 kg/m^2^), and parity (nulliparous (0) and parous (>0)). Race and ethnicity were self-identified by patients upon registration for encounters within our medical system. An area-level neighborhood deprivation index from 2015 was also included [20]. The neighborhood deprivation index is a census tract-level index which measures material deprivation in communities across the country. It is comprised of six components, including median household income, the proportion of vacant homes, and the proportion of the population with the following indicators: annual incomes below the poverty level, less than a high school education, no health insurance, and receiving public assistance income. The deprivation index ranges from 0 to 1, with values closer to 0 referring to the least deprived communities and values closer to 1 referring to more deprived communities.

### 2.5. Statistical Analysis

To quantify associations of walkability with PTB, as well as with sPTB and mPTB, compared to term births, we built multilevel logistic and multinomial logistic regression models, respectively, to calculate odds ratios (ORs) and 95% confidence intervals (CIs) per 10-point increment increases in Walk Score rankings. In adjusted models, we used a random effect for the census tract to address shared variance of neighborhood factors. Models were adjusted for categorical age and pre-pregnancy BMI, insurance, parity, and continuous neighborhood deprivation index. We also performed a post-hoc analysis of walkability with specific indications for mPTB.

A secondary analysis was conducted stratified by race/ethnicity, rather than adjusting for race/ethnicity to observe whether associations of walkability with PTB, sPTB, and mPTB varied by race/ethnicity. We also included race/ethnicity × walkability interaction terms in unstratified adjusted models. We performed these analyses because it is conceivable that effects of walkability among individuals due to co-exposures colinear with race (i.e., exposure to racism) may modify patients’ responses to area-level exposures. Additionally, walkability may vary in its effectiveness in different neighborhoods, which may also vary with respect to racial/ethnic composition.

We also performed a post-hoc causal mediation analysis to quantify the extent to which neighborhood walkability may explain Black-White disparities in mPTB phenotypes. This model was adjusted for age, pre-pregnancy BMI, insurance, parity, and neighborhood deprivation index. Analyses were carried out using SAS 9.4, Cary, NC, except for mediation, which was performed using the *mediation* package [21] in R version 4.0.5.

## 3. Results

### 3.1. Study Population

There were 19,203 individuals included in the analytic dataset (Figure 2). Table 1 displays characteristics of patients in the GeoBirth cohort. The majority of patients were 25–34 years old, 53.1% self-identified as Non-Hispanic Black, 27.8% as Non-Hispanic White, 6.2% as Asian, 8.4% as Hispanic, and 4.5% as multiracial, other, or unspecified. Additionally, 56.1% were on public, self-pay, or other forms of insurance. 

#### 3.1.1. Bivariate Associations of Demographics with Walkability

Patient characteristics associated with higher neighborhood walkability included age > 35 years, Non-Hispanic White and Asian racial identities, private insurance, BMI < 25 kg/m^2^, and nulliparity (Table 2).

#### 3.1.2. Bivariate Associations with PTB

Compared to patients 25-<35 years of age, younger (<25 years) and older patients (≥35 years) patients were more likely to have sPTB and mPTB, respectively. Patients who self-identified as Non-Hispanic Black, obese patients, and those with public insurance were also more likely to have PTB, sPTB, and mPTB (Table 1).

Bivariate associations of walkability with PTB and sPTB were not significant. However, patients who had a mPTB were more likely to live in census tracts with lower walkability compared to patients who had a term birth (*p* = 0.0012) (Table 1).

#### 3.1.3. Adjusted Associations of Walkability with PTB

In models adjusted for maternal age, BMI, insurance status, parity, and neighborhood deprivation, we did not detect associations of walkability with PTB or sPTB (Table 3). However, higher walkability was associated with lower odds of mPTB (aOR:0.90, 95% CI: 0.83–0.98) (Table 3). Furthermore, we found that walkability was associated with significantly lower odds of mPTB, specifically due to hypertensive disorders of pregnancy (HDP) in an unadjusted model (OR 0.88, 95% CI: 0.79–0.98), with similar but not significant results in the adjusted model (aOR 0.91, 95% CI: 0.80–1.03) (Appendix A Table A1).

When stratifying by race/ethnicity, higher walkability was not significantly associated with PTB, sPTB, or mPTB for any group (Table 3). While associations were not statistically significant, the direction of the association between higher walkability and mPTB differed for Non-Hispanic White (aOR: 0.87, 95% CI: 0.75, 1.01) and Non-Hispanic Black patients (aOR: 1.05, 95% CI: 0.92, 1.21) (Table 3). In a model with both Black and White patients, there was a significant interaction between race and walkability on the outcome of mPTB (*p* = 0.03). Additionally, in the post-hoc analysis, walkability was associated with lower odds of mPTB due to HDP among White patients (aOR 0.74, 95% CI: 0.62–0.89), but higher odds among Black patients (aOR 1.24, 95% CI: 1.01, 1.52) (interaction *p* = 0.0005) (Figure 3). For all other racial and ethnic groups, as well as for PTB and sPTB, no other associations with walkability were significant (Appendix A Table A1). Finally, our causal mediation analysis did not reveal significant mediation of Black-White disparities in mPTB by walkability (5.0%, *p* = 0.36) or for mPTB due to HDP (2.1%, *p* = 0.68).

## 4. Discussion

In a Philadelphia hospital system-based birth cohort study, we found that higher walkability was associated with lower odds of mPTB, but not sPTB. Furthermore, we found a significant interaction between race/ethnicity and walkability on the outcome of mPTB, indicating that higher walkability may be more protective for Non-Hispanic White than Non-Hispanic Black patients.

In one previous North Carolina study by Vinikoor-Imler et al., a more walkable neighborhood was associated with a lower risk of PTB among White patients, yet the effects of walkability on specifically mPTB and sPTB were not reported [10]. Our findings differed from those findings; we did not detect significantly lower odds of PTB overall or among White patients. However, the investigators did report that walkability was associated with lower odds of pregnancy-induced preterm hypertension, which is consistent with our study. We found that higher walkability was associated with lower odds of mPTB, and specifically, mPTB due to HDP. Additionally, we detected a significant interaction between race and walkability, suggesting that, like in North Carolina, walkability may be more protective for mPTB, and specifically mPTB for HDP, among White compared to Black patients.

Since HDP and placental insufficiency that leads to impaired fetal growth are the two most common reasons for mPTB [5], it is possible that walkability leads to improved cardiovascular function and lower rates of mPTB. While we did not measure patients’ physical activity, we speculate that walkability could be protective for mPTB by encouraging more exercise; individuals living in unwalkable neighborhoods may be less physically active [9]. According to the American College of Obstetricians and Gynecologists, physical activity can aid in preventing HDP and gestational diabetes, both of which can increase the risk of mPTB [22]. A 2017 systematic review and meta-analysis conducted by Magro-Malosso et al. found that aerobic exercise for about 30–60 min two to seven times per week during pregnancy is associated with a significantly reduced risk of HDP, and gestational hypertension specifically [23].

It is not clear why the association between walkability and mPTB differed by race/ethnicity in our study. However, it is possible that higher walkability may not equally encourage physical activity across segregated neighborhoods. Walkability, which was measured using Walk Score rankings, was determined by assigning points depending on the walking distance to amenities (e.g. grocery stores, parks, libraries, drug stores, and clothing stores) [24] and the ability to accomplish daily errands through walking. Residential racial and economic segregation may have led to differential quality of amenities. The availability and access to quality amenities like grocery stores, healthcare facilities, public safety, education, etc., are influenced by residential segregation [25]. For example, the United States Department of Agriculture (USDA) released a report to Congress on the factors that contribute to an individual’s ability to access nutritious and affordable food. They found that specifically in urban core areas, income inequality and racial segregation are the largest contributing factors to an individual’s ability to access healthier foods [26]. In a neighborhood, though there may be amenities nearby, like small convenience stores that factor into the Walk Score ranking, these small convenience stores tend to lack healthier options, which in turn can negatively impact overall diet and health [27]. Other studies reiterate that though there may be amenities around, which would be calculated within the Walk Score ranking, Walk Score ranking does not factor in the quality of these amenities [28]. Since higher walkability was not associated with protective odds of mPTB among Black patients, it is possible that improving the quality of amenities in these census tracts could alter the association between walkability and mPTB.

It is also possible that even though a neighborhood may be “walkable,” residents may not actually walk throughout the neighborhood. Neighborhood violence, which is more prevalent in neighborhoods of color [29] and other sociocultural factors, may diminish the potential health benefits of walkability if they preclude walking itself. A 2007 study was conducted by Bennet et al., in 12 urban low-income housing complexes in Boston, which comprised mostly of minoritized racial/ethnic residents [30]. Participants self-reported perceptions of neighborhood safety and confidence in their ability to be physically active while they also simultaneously reported data from their pedometer. Participants who reported feeling unsafe at night had significantly fewer steps per day. The investigators also found that generally perceiving their neighborhood as unsafe was associated with significantly lower odds of high physical activity self-efficacy in both men and women [30]. Therefore, though the walkability of a neighborhood may be high, that may not translate into increased walking habits. Moreover, with respect to PTB, a study in Detroit, Michigan was conducted to determine if psychosocial factors like depressive symptoms, psychological distress, and perceived stress mediated the associations between perceived neighborhood factors (including neighborhood safety and walkability) and PTB among Black participants. Though the cohort was small, with only 399 participants, the investigators found that residing in less safe neighborhoods was associated with depressive symptoms which, in turn, was associated with PTB. However, the association between perceived walkability and PTB was not mediated by psychosocial factors [31]. Nonetheless, other neighborhood factors, like perceived neighborhood safety, may not only prevent an individual from walking, but may also be associated with psychological distress or depressive symptoms that could contribute to conditions that lead to mPTB for Black patients.

### Strengths and Limitations

A major strength of our study was that we rigorously phenotyped sPTB and mPTB, a step that most environmental epidemiologic studies do not take. Furthermore, our cohort was diverse, and we accounted for both individual- and area-level confounders, including neighborhood deprivation. Including neighborhood deprivation in our models is particularly important since walkability may be confounded by other neighborhood socioeconomic variables. Furthermore, unlike the Centers for Disease Control and Prevention Social Vulnerability Index and the more commonly used Area Deprivation Index [32,33], the neighborhood deprivation index we used does not incorporate residential racial and ethnic composition in its calculation, which is important when analyzing potential contributors to individual-level racial disparities in health. Limitations of our study include a lack of interaction with patients to garner information on walking habits. Additionally, like all observation studies, findings could be the result of unmeasured confounding variables. Walk Score rankings were ascertained in 2021 (and are updated every 6 months). As such, the exposures were extrapolated back in time. However, walkability is relatively stable over time; an analysis of seven US metropolitan cities from 2000–2010 showed minimal changes to walking destinations [34]. Finally, we performed multiple comparisons, and as such, our finding of an association of walkability with mPTB may have been due to Type-I error. However, it is plausible that the association is real given the potential cardiovascular benefits of walking that could improve placental function and overall pregnancy health. Despite the noted limitations, our study highlights the importance of phenotyping PTB when considering area-level risk factors and identifying potential interventions to reduce the risks of sPTB and mPTB.

## 5. Conclusions

In conclusion, we found that walkability may be associated with lower odds of mPTB, but that benefits may not be equal across populations and neighborhoods. Higher walkability was associated with non-significantly higher odds of mPTB among Black and lower odds of mPTB among White patients. A deeper understanding of neighborhood determinants of physical activity and pregnancy health may be critical to reducing the risk of PTB and reducing racial disparities in PTB in urban settings.

## Figures and Tables

**Figure 1 ijerph-20-05932-f001:**
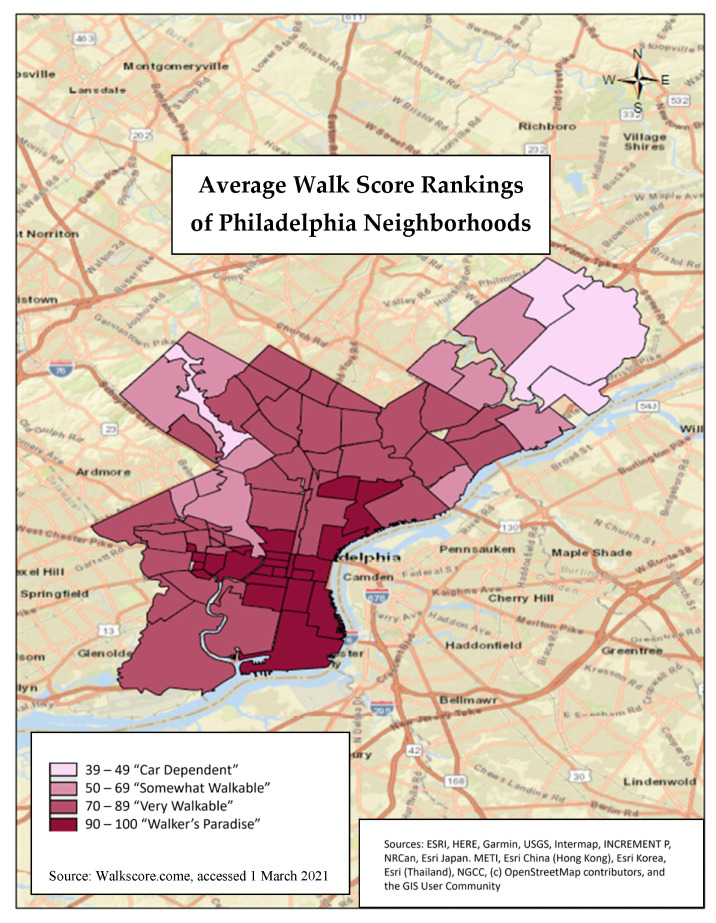
Average Walk Score rankings of Philadelphia neighborhoods listed on WalkScore.com (accessed on 1 March 2021). The Walk Score ranking is calculated using a point system. Points are determined by distance to amenities and the ability to conduct daily errands.

**Figure 2 ijerph-20-05932-f002:**
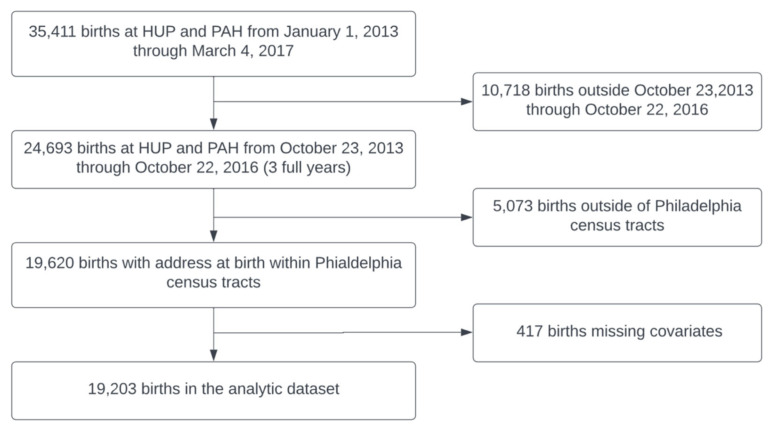
GeoBirth participants included in the analytic cohort; births occurred at the Hospital of the University of Pennsylvania (HUP) and Pennsylvania Hospital (PAH).

**Figure 3 ijerph-20-05932-f003:**
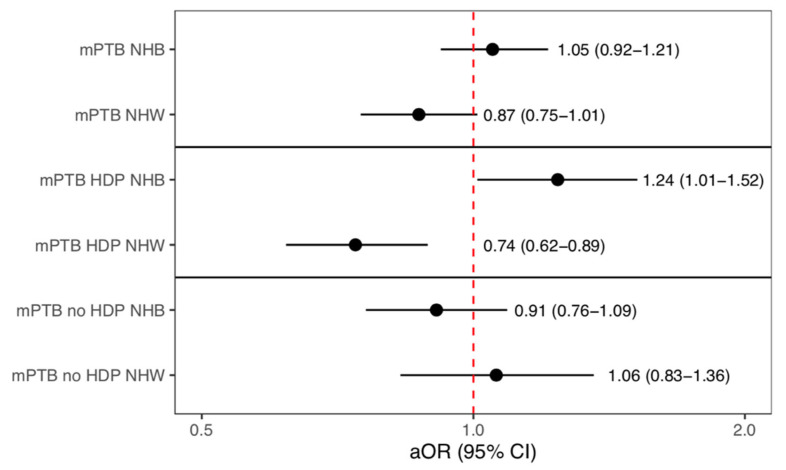
Adjusted models stratified by race/ethnicity ^a^ showing the association with walkability and medically indicated preterm birth (mPTB), mPTB due to hypertensive disorders **^†^**, and mPTB due to other reasons. ^a^ Non-Hispanic Black (NHB), Non-Hispanic White (NHW), **^†^** Hypertensive Disorders of Pregnancy (HDP).

**Table 1 ijerph-20-05932-t001:** Characteristics of patients residing in Philadelphia who gave birth from 23 October 2013 through 22 October 2016 (n = 19,203) by preterm birth (PTB), spontaneous PTB (sPTB), and medically indicated PTB (mPTB).

	Total (n = 19,203)	Term (n = 17,477)	PTB (n = 1726)		sPTB (n = 1070)		mPTB (n = 656)	
Charcteristics	n (%)	n (%)	n (%)	*p* ^a^	n (%)	*p* ^a^	n (%)	*p* ^a^
**Age (years)**				0.0275 *		<0.0001 *		0.0133 *
<25	5213 (27.1)	4700 (26.9)	513 (29.7)		361 (33.7)		152 (23.2)	
25-<35	10,579 (55.1)	9675 (55.4)	904 (52.4)		542 (50.7)		362 (55.2)	
≥35	3411 (17.8)	3102 (17.7)	309 (17.9)		167 (15.6)		142 (21.6)	
**Race/ethnicity ^b^**				<0.0001 *		<0.0001 *		<0.0001 *
Non-Hispanic White	5344 (27.8)	5018 (28.7)	326 (18.9)		216 (20.2)		110 (16.8)	
Non-Hispanic Black	10,195 (53.1)	9065 (51.9)	1130 (65.5)		667 (62.3)		463 (70.6)	
Asian	1191 (6.2)	1112 (6.4)	79 (4.6)		64 (6.0)		15 (2.3)	
Hispanic	1618 (8.4)	1493 (8.5)	125 (7.2)		79 (7.4)		46 (7.0)	
Multiracial, other, unknown	855 (4.5)	789 (4.5)	66 (3.8)		44 (4.1)		22 (3.4)	
**Insurance status**				<0.0001 *		<0.0001 *		<0.0001 *
Private	8432 (43.9)	7835 (44.8)	597 (34.6)		371 (34.7)		226 (34.5)	
Public/self-pay/other	10,771 (56.1)	9642 (55.2)	1129 (65.4)		699 (65.3)		430 (65.5)	
**Body mass index (kg/m^2^)**				<0.0001 *		0.0006 *		<0.0001 *
<25	7733 (40.3)	7144 (40.9)	589 (34.1)		401 (37.5)		188 (28.7)	
≥25-<30	3719 (19.4)	3392 (19.4)	327 (19.0)		190 (17.8)		137 (20.9)	
≥30	4256 (22.2)	3776 (21.6)	480 (27.8)		233 (21.8)		247 (37.7)	
Missing	3495 (18.2)	3165 (18.1)	330 (19.1)		246 (23.0)		84 (12.8)	
**Parity**				0.0120 *		0.0221 *		0.2165
0	8494 (44.2)	7780 (44.5)	714 (41.4)		438 (40.9)		276 (42.1)	
>0	10709 (55.8)	9697 (55.5)	1012 (58.6)		632 (59.1)		380 (57.9)	
**Walk Score ranking ^†^, mean (SD)**	84.9 (9.6)	85.0 (9.7)	84.5 (8.5)	0.0450 *	84.9 (8.2)	0.9476	83.7 (9.0)	0.0012 *
**Neighborhood deprivation index ^, mean (SD)**	0.49 (0.14)	0.49 (0.14)	0.51 (0.13)	<0.0001 *	0.51 (0.13)	<0.0001 *	0.52 (0.13)	<0.0001 *

^a^ *p* value compared to terms using Chi-square or *t*-test as appropriate. ^b^ Race/ethnicity self-identified during patient registration. ^†^ Walk Score ranking ranges: 0–24 (all errands are car-dependent), 25–49 (most errands require a car), 50–69 (some errands can be done on foot), 70–89 (most errands can be done on foot), 90–100 (daily errands do not require a car). ^ Neighborhood deprivation index ranges from 0–1, with 0 least deprived and 1 most deprived (includes census tract poverty, education, and vacant housing indicators). * Significant results.

**Table 2 ijerph-20-05932-t002:** Walk Score ranking mean and standard deviation by characteristics of patients residing in Philadelphia who gave birth from 23 October 2013 through 22 October 2016 (n = 19,203).

	Walk Score Ranking ^†^	*p* ^a^
	Mean (SD)	
**Age (years)**		<0.0001 *
<25	83.5 (7.4)	
25-<35	85.1 (10.1)	
≥ 35	86.6 (10.6)	
**Race/ethnicity ^b^**		<0.0001 *
Non-Hispanic White	88.2 (11.6)	
Non-Hispanic Black	82.5 (7.3)	
Asian	88.2 (10.6)	
Hispanic	87.2 (8.6)	
Multiracial, other, unknown	84.7 (11.7)	
**Insurance status**		<0.0001 *
Private	86.4 (11.4)	
Public/self-pay/other	83.8 (7.6)	
**Body mass index (kg/m^2^)**		<0.0001 *
<25	86.7 (10.3)	
≥25-<30	84.4 (9.7)	
≥30	83.0 (8.6)	
Missing	83.9 (8.3)	
**Parity**		<0.0001 *
0	85.7 (10.2)	
>0	84.3 (9.0)	

^a^ *p* value from ANOVA or *t*-test. ^b^ Race/ethnicity self-identified during patient registration. ^†^ Walk Score ranking ranges: 0–24 (all errands are car-dependent), 25–49 (most errands require a car), 50–69 (some errands can be done on foot), 70–89 (most errands can be done on foot), 90–100 (daily errands do not require a car). * Significant results.

**Table 3 ijerph-20-05932-t003:** Unadjusted and adjusted ^a^ odds ratios (OR) of preterm birth (PTB), spontaneous preterm birth (sPTB), and medically indicated preterm birth (mPTB) per 10-point increase in Walk Score ranking **^†^** in all patients and then in models stratified by race/ethnicity.

	Walk Score Ranking ^†^ Mean (SD)	Unadjusted OR (95% CI)	Adjusted OR ^a^ (95% CI)	Interaction *p*-Value
**All**				
Term	85.0 (9.7)	Ref.	Ref.	
PTB	84.5 (8.5)	0.96 (0.92, 1.00)	0.99 (0.94, 1.03)	-
sPTB	84.9 (8.2)	1.02 (0.94, 1.09)	1.04 (0.97, 1.12)	-
mPTB	83.7 (9.0)	0.88 (0.82, 0.95) *	0.90 (0.83, 0.98) *	-
**Asian, Hispanic, other, multiracial, and unknown**				
Term	86.9 (10.2)	Ref.	Ref.	
PTB	87.2 (9.4)	1.03 (0.91, 1.15)	1.06 (0.94, 1.18)	0.45
sPTB	87.4 (8.6)	1.05 (0.90, 1.22)	1.10 (0.93, 1.29)	0.87
mPTB	86.7 (11.0)	0.97 (0.78, 1.22)	0.99 (0.79, 1.24)	0.25
**Non-Hispanic Black**				
Term	82.4 (7.3)	Ref.	Ref.	
PTB	82.8 (6.8)	0.98 (0.91, 1.05)	0.99 (0.91, 1.08)	0.16
sPTB	82.9 (6.6)	1.11 (0.99, 1.25)	1.09 (0.96, 1.22)	0.97
mPTB	82.7 (6.9)	1.06 (0.93, 1.21)	1.05 (0.92, 1.21)	0.03 *
**Non-Hispanic White**				
Term	88.3 (11.6)	Ref.	Ref.	
PTB	87.9 (11.4)	0.98 (0.91, 1.05)	0.99 (0.91, 1.08)	Ref.
sPTB	89.1 (10.1)	1.07 (0.94, 1.22)	1.08 (0.94, 1.24)	Ref.
mPTB	85.7 (13.3)	0.85 (0.74, 0.98) *	0.87 (0.75, 1.01)	Ref.

^a^ Adjusted for age, parity, pre-pregnancy body mass index, insurance, and neighborhood deprivation index. **^†^** Walk Score ranking ranges: 0–24 (all errands are car-dependent), 25–49 (most errands require a car), 50–69 (some errands can be done on foot), 70–89 (most errands can be done on foot), 90–100 (daily errands do not require a car). * Significant results.

## Data Availability

Data for Walk Score are available here: https://www.walkscore.com/PA/Philadelphia (accessed for this analysis 1 March 2021) Access to identified *GeoBirth* data are restricted because they are derived from a hospital system. Requests to collaborate should be made to Dr. Burris burrish@chop.edu.

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
