# Peer review of "Neighborhood Walkability as a Risk Factor for Preterm Birth Phenotypes in Two Philadelphia Hospitals from 2013–2016"

_ijerph, 2023, doi:10.3390/ijerph20115932_

Round 1
Reviewer 1 Report
The authors have presented findings on the association between walkability score and pre-term birth. Overall, the manuscript is well presented. I have some minor suggestions for the authors to consider: Outcome variables: Authors will need to definition they used to classify the PTB, sPTB and mPTB. Medically indicated PTB may have certain medical conditions therefore, mentioning here will increase the transparency of their classification of sPTB vs mPTB. Defining other variables: Information such as details on the factors age (<25; 25-<35; and ≥35 years) and pre-pregnancy BMI (<25; 150 25-<30; and ≥30 kg/m2 ), insurance (public compared to private), parity (nulliparous (0) and 151 parous (>0)), and continuous neighborhood deprivation index must be explained in details in the definition of variable section. It is important in this case as it is highly context specific. Once it is done, the information does not need to be repeated in the statistical analyses section. Presentation of findings: 3.1.1 to 3.1.3- Please re-structure the presentation. Currently, it is difficult to ready. An alternative approach could be, briefly summarizing what are significant in the unadjusted model, and then going with the final and alternative models (eg. using interaction terms). Presentation of the tables: The table has a large volume of data in the models. Therefore, I request authors indicate the significant differences using * so the reader can easily navigate the table. Table headings need to the top of the table, not inside the table. Figure 3- please use a footnote to indicate what abbreviations mean so it is easier to follow your findings.Author Response
Please see the attachment

Reviewer 2 Report
Introduction: The introduction is well written with the background information, research problem, research objective(s), and a proper description of the research question(s). The authors clearly state the problem being investigated and summarize the relevant research.
Methodology: The methods are clearly described to outline the experiments conducted and the authors provide the necessary information with relevance to data collection approach and data analysis technique from these studies.
Results The results are carefully laid out in a logical sequence and the appropriate analysis has been conducted. The interpretation of the results are carefully discussed with relevance to the research conclusions.
Discussion: The discussion section is relevant to claims that are supported by the data
References: The authors provide an adequate literature review and the manuscript builds upon previous research that is referenced appropriately.
Tables / Figures: The tables are well presented in this manuscript.
The article can be published with minor revisions.
Please mention the weak and strong points of your study
